Establishment and validation of systemic inflammatory index model and risk assessment of PVT in cirrhosis after splenectomy—a retrospective study

Deng Xin 1
Liao Wenyan 2
Jiang Xinmiao 1
Tu Shun 1
Xie Xiangmin 1
Xiao Yuji 1
Chen Wuyao 1
Zeng Huan 2
Ding Chengming dingchengming83@163.com 1
1 The First Affiliated Hospital, Department of Hepatopancreatobiliary Surgery, Hengyang Medical School, University of South China , Hengyang , Hunan , China
2 The First Affiliated Hospital, Department of Gynaecology and Obstetrics, Hengyang Medical School, University of South China , Hengyang , Hunan , China
Fernandes Carlos Eurico
Electronic publication date: 2025 May 12
Publication date: 2025
Volume: 13
Electronic Location ID: e19254
Received 2024 Jun 21; Accepted 2025 Mar 12
Copyright: ©2025 Deng et al.
Copyright year: 2025
Copyright holder: Deng et al.
License: This is an open access article distributed under the terms of the Creative Commons Attribution License, which permits unrestricted use, distribution, reproduction and adaptation in any medium and for any purpose provided that it is properly attributed. For attribution, the original author(s), title, publication source (PeerJ) and either DOI or URL of the article must be cited.
License URL: https://creativecommons.org/licenses/by/4.0/

Keywords: Splenectomy, Portal vein thrombosis, Cirrhosis, Systemic inflammation, Practical model

Funding: Hunan Provincial Natural Science Fund No. 2022JJ30530 2022JJ30526 Scientific Research Fund Project of Hunan Provincial Health Commission No. 202105010225 Clinical Medical Technology Innovation Guidance Project from Department of Science and Technology of Hunan Province No. 2021SK51812 2021SK51802 Science and Technology Research Project (Basic Applied Research Program) Department of Science and Technology of Hengyang City No. 202150063929 This work was supported by Hunan Provincial Natural Science Fund (No. 2022JJ30530, 2022JJ30526), Scientific Research Fund Project of Hunan Provincial Health Commission (No. 202105010225), Clinical Medical Technology Innovation Guidance Project from Department of Science and Technology of Hunan Province (No. 2021SK51812, 2021SK51802) and Science and Technology Research Project (Basic Applied Research Program) from Department of Science and Technology of Hengyang City (No. 202150063929). The funders had no role in study design, data collection and analysis, decision to publish, or preparation of the manuscript.

==============================
Objective

The study aimed to create and validate a straightforward nomogram to predict portal vein thrombosis (PVT) in cirrhotic patient post-splenectomy, and investigate the predictive potential of systemic inflammation markers. One objective of the study was to develop a predictive model utilizing these markers to detect high-risk individuals early on.

Methods

A retrospective analysis was conducted on 184 cases of patients with cirrhosis who underwent splenectomy at The First Affiliated Hospital of University of South China from January 2015 to September 2023. The cohort was randomly divided into training (n = 130) and validation (n = 54) groups. Univariate and multivariate logistic regression analysis was employed to construct the prediction model. The performance of the nomogram was evaluated based on its ability to discriminate, calibrate, and demonstrate clinical utility.

Results

According to univariate and multivariate logistic regression analysis, we found six prediction indexes of PVT in patients with cirrhosis after splenectomy: postoperative neutrophil-to-lymphocyte ratio (NLR), postoperative derived NLR (dNLR), C-reactive protein to albumin ratio (CAR), portal vein diameter (DPV), platelet change value (PVB), and D-dimer (p-value < 0.05). Our clinical prediction model was created based on the aforementioned risk factors and demonstrated superior predictive power in both the primary cohort (AUC = 0.876) and validation cohort (AUC = 0.817). The calibration curve demonstrated satisfactory agreement between model predictions and actual observations, and the decision curve analysis (DCA) curve indicated high clinical net benefit.

Conclusion

Postoperative NLR, dNLR, CAR, PVB, DPV, and D-dimer were identified as the independent risk factors of PVT in cirrhotic patients post splenectomy. We had successfully established and validated a novel predictive model with good performance, based on systemic inflammatory indices in predicting PVT in cirrhosis after splenectomy.

Introduction

Cirrhosis was a globally prevalent and serious liver disease with high morbidity and mortality rates. Annually, approximately 2 million individuals succumb to liver disease worldwide, with 1 million deaths attributed to cirrhosis, viral hepatitis, and hepatocellular carcinoma combined. Over 60% of liver disease-related deaths occur in men, with cirrhosis ranking as the 11th most common cause of death and the third leading cause of death among those aged 45–64 years (Zhou et al., 2019). The progression of liver cirrhosis gives rise to severe complications such as ascites, hypersplenism, and portal hypertension , with hypersplenism being a significant pathophysiological characteristic of liver cirrhosis (Jeker, 2013; McCormick & Murphy, 2000). To address hypersplenism-related issues, splenectomy, the removal or partial removal of the spleen, is often recommended by clinicians to ameliorate clinical symptoms (Weledji, 2014; Zhu et al., 2023). However, it is reported that splenectomy could increased risk of portal vein thrombosis (PVT) in liver cirrhosis patients (Qi et al., 2016; Jiang et al., 2016). Therefore, PVT is a noteworthy complication associated with splenectomy in PVT.

PVT is generally defined as thrombosis in the main portal vein or its branches, including the mesenteric vein and splenic vein, with or without involvement of the splenic vein or superior mesenteric vein. The reported incidence of PVT after splenectomy ranges from 9.8% to 47.9% (Lin et al., 2023). Severe PVT can lead to ascites, exacerbating liver damage and potentially resulting in liver failure in cirrhosis patients. Moreover, gastrointestinal tract congestion, bacterial translocation, and endotoxin release may cause extensive intestinal necrosis. Notably, a study revealed that 67% of PVT cases were asymptomatic when screened via enhanced computed tomography (CT), suggesting an underestimated incidence of this complication (Ruiz-Tovar & Priego, 2017). While anticoagulation is a crucial method for preventing and treating PVT post-splenectomy, consensus on risk factors and the optimal timing for postoperative antithrombotic drugs remains elusive. Therefore, it is imperative to establish preventive measures promptly after splenectomy for patients with hypersplenism. Specific factors associated with PVT after splenectomy include portal vein diameter (DPV), splenic vein diameter (DSV), portal vein flow velocity, spleen volume, plasma D-dimer level, postoperative platelet count, and postoperative platelet count elevation rate (Lin et al., 2023; Li et al., 2022). Surgery-induced stress response, including inflammatory stress response, is a well-known consequence of any bodily injury, as highlighted by Alazawi’s comprehensive review in 2015 (Alazawi et al., 2016). This stress response is positively correlated with the degree of surgical trauma, as demonstrated by a meta-analysis covering 164 studies and 14,362 patients. Systemic inflammation markers, a group of widely utilized biomarkers, play a pivotal role in assessing inflammatory status for diagnosing, evaluating disease severity, monitoring treatment effects, and predicting disease prognosis in clinical medicine. It is reported that systemic inflammation as a risk factor for portal vein thrombosis in cirrhosis (Nery et al., 2021). Carr, Guerra & Donghia (2020) found that blood levels of neutrophil count/lymphocyte count (NLR), platelet count/lymphocyte count (PLR), erythrocyte sedimentation rate (ESR), C-reactive protein (CRP), alpha-fetoprotein (AFP) and gamma-glutamyl transpeptidase (GGTP) were significantly related to the presence of PVT. The underlying mechanisms may be reported that inflammatory factors can cause increased platelet reactivity, damage vascular endothelium, then lead to blood stasis, and cause thrombosis (Zhou et al., 2023). All of these confirm that systemic inflammation factors are related with portal vein thrombosis in cirrhosis. Given the approximately 9.8%–47.9% overall incidence rate of PVT after splenectomy, this study explores the disrupted inflammatory response following splenectomy and its potential association with postoperative PVT. Utilizing systemic inflammation indicators, the study aims to construct a novel risk prediction model to assess PVT risk in patients with post-hepatitis cirrhosis undergoing splenectomy. This model seeks to offer clinicians a more accurate risk assessment tool, thereby facilitating personalized medical care, reducing adverse outcomes related to PVT, and improving the overall quality of life for patients. The research holds significant clinical relevance in enhancing survival rates, quality of life, and preventing severe complications in cirrhosis patients.

Methods and Materials

Research group

We conducted a retrospective analysis involving 528 patients diagnosed with post-hepatitis cirrhosis and hypersplenism who underwent splenectomy at our hospital from January 2015 and September 2023. Subsequent follow-up care was provided for all patients at the same institution. For this study, PVT was defined as either complete or partial thrombotic occlusion affecting the main portal vein, its branches, and the superior mesenteric vein (Wu et al., 2022). The criteria for splenectomy in patients with liver cirrhosis included meeting one of the following conditions: white blood cells < 3 ×109/L and/or platelets < 30 ×109/L; splenomegaly of grade II with upper gastrointestinal bleeding or clear signs of spleen hyperactivity; or grade III or above splenomegaly (Zhou et al., 2023; Wu et al., 2022). The inclusion criteria for cases were: (1) patients aged 18 years and above; (2) clinically confirmed causes of liver cirrhosis, including viral hepatitis A, B, C, alcoholic hepatitis, autoimmune hepatitis, etc., combined with hypersplenism; (3) clear indications for splenectomy; and complete preoperative and postoperative baseline evaluation data, including peripheral blood and imaging. Exclusion criteria encompassed: (1) patients undergoing malignant tumor surgery combined with spleen and cholecystectomy; (2) those with benign and malignant space-occupying lesions of the spleen; (3) individuals experiencing traumatic splenic rupture; (4) patients with a history of long-term hormone therapy and blood diseases; (5) cases requiring laparoscopic conversion to laparotomy; (6) individuals with severe dysfunction of critical organs such as the heart, brain, or kidneys, rendering them unable to tolerate surgery or experiencing coagulation dysfunction; (7) instances of preoperative PVT; (8) previous splenic embolization or anticoagulant use; and (9) incomplete clinical data. Ultimately, a total of 184 patients with hepatitis-related cirrhosis and spleen function met the criteria for inclusion in this study. The patient selection and study design are visually presented in Fig. 1. Ethical approval for this study was obtained from the Medical Ethics Committee of The First Affiliated Hospital of the University of South China (NO. 2023ll1017001). And we received a waiver of the need for informed consent from participants of our study.

Figure 1 Research flow chart.

Data collection

The clinical attributes of all patients were retrieved from the electronic health records system of our hospital. This information primarily encompasses gender, age, body mass index, previous treatment records, medical history, intraoperative and postoperative clinical details, blood biochemical markers, and imaging data. Blood biochemical indicators consist of blood routine parameters, including white blood cells (WBC) with a reference interval of 3.5∼9.5 ×109/L, neutrophil count (N) with a reference interval of 1.8∼6.30 × 109/L, lymphocyte count (L) with a reference interval of 1.1∼3.20 × 109/L, platelet count with a reference interval of 125.0∼350.0 × 109/L, prothrombin time (PT) with a reference interval of 9.0 ∼14.5 s, activated partial coagulation time (APTT) with a reference interval of 22.0 ∼43.0 s, international normalized ratio (INR) with a reference interval of 0.80 ∼1.30, D-dimer with a reference interval of 0∼1 mg/L, procalcitonin (PCT) with a reference interval of 0∼0.05 ng/ml, interleukin (IL-6) with a reference interval of < 7 pg/ml, and C-reactive protein (CRP) with a reference interval of 0∼5 mg/L. Calculations were performed for neutrophil count/lymphocyte count (NLR), neutrophils/whole blood white blood cell count - neutrophil count (dNLR), platelet count/lymphocyte count (PLR), lymphocyte count/monocyte count (LMR), C-reactive protein (mg/L) / albumin (g/L) (CAR), systemic immune inflammatory index (SII) (platelets x neutrophils / lymphocyte count), fibrinogen (g/L) / albumin (g/L) (FAR), and platelet change value (PVB), representing the change in the mean platelet count one week post-surgery compared to the preoperative count. The preoperative Child-Pugh classification was divided into three levels: A, B, and C (Jelic, Sotiropoulos & ESMO Guidelines Working Group, 2010), and the class C was excluded in our study. Additionally, preoperative data such as spleen thickness (SPT), portal vein diameter (DPV), and splenic vein diameter (DSV) were collected. In Fig. 2, we present a correlation heatmap illustrating all candidate variables. The size and color of the circles in the correlation matrix signify the correlation strength between the respective variables. Darker shades of blue indicate a stronger positive correlation, while darker red shades signify a stronger negative correlation between variables.

Figure 2 Correlation matrix between all candidate variables.

Variable definitions

In this investigation, postoperative contrast-enhanced CT served as the primary diagnostic tool for detecting PVT (Intagliata, Caldwell & Tripodi, 2019). Additionally, color Doppler ultrasonography was utilized as a supplementary diagnostic method (Tran et al., 2010). Routine abdominal ultrasound and contrast-enhanced CT examinations were conducted within one week before surgery. Furthermore, routine ultrasound and CT scans were performed from 3 to 5 days post-surgery, or when there were clinical suspicions of PVT, marked by symptoms such as postoperative fever, severe abdominal pain, vomiting, and abnormal liver function, as well as leukocytosis (Harding et al., 2015; Stotts, Wentworth & Northup, 2021; Pan et al., 2022).

Splenic thickness (SPT) was defined as the vertical distance from the splenic hilum to the lateral edge cutting point. Portal vein diameter (DPV) was measured as the maximum anteroposterior diameter at the junction with the hepatic artery when the patient holds their breath (Lehmann et al., 2008). The diameter of the splenic vein (DSV) was defined as the maximum diameter of the splenic vein, and the platelet change value (PVB) was defined as the change in the arithmetic mean number of platelets within one week after surgery compared with the preoperative platelet count.

Model development

All candidates were randomly divided into two groups, with 70% assigned to the training cohort and 30% to the validation cohort. Model development took place exclusively in the training cohort. Multiple stepwise logistic regression analysis, conducted using SPSS Statistics 25, was employed to identify valuable variables.

Model verification

The stability and clinical utility of the model were assessed through curves, calibration curves, and decision curve analysis (DCA). The area under the curve (AUC) from ROC curves estimated the recognition rate of each model using the “pROC” R package. Calibration curves, assessed using the “Resource-Selection” R package, were employed to assess the calibration ability of each model, calibrated with 1000 bootstrap samples to reduce overfitting bias. The clinical applicability of the model was further assessed by DCA using the “rms” and “rmda” R packages.

Statistical analysis

IBM SPSS Statistics 25 and R statistical software (version 4.3.1, https://www.r-project.org) were utilized for statistical analysis and model evaluation. To mitigate possible nonlinear effects, all continuous variables were dichotomized after determining the optimal cutoff value from the ROC curve (Xia et al., 2020). Categorical variables were described as numbers (percentages) and compared using the chi-square test or Fisher’s exact test as appropriate. The Spearman correlation coefficient was employed to determine the correlation between candidate variables. All statistical tests were two-tailed, and significance was set at P < 0.05.

Results

Demographics and characteristics of enrolled patients

Following meticulous screening based on the predefined inclusion and exclusion criteria, a total of 184 post-splenectomy patients with post-hepatitis cirrhosis and hypersplenism were enrolled in our conclusive study cohort. The participants were randomly and automatically divided into a training cohort (n = 130, 70%) and a validation cohort (n = 54, 30%). The incidence of PVT was observed to be 30.8% in the training group and 29.6% in the independent validation group. Specifically, in the training cohort and validation cohort, 40 cases (30.8%) and 16 cases (29.6%) were diagnosed with PVT, respectively, resulting in an overall incidence of PVT of 30.4%. A comprehensive overview of all data, encompassing patient demographics, clinical details, and laboratory characteristics, is presented in Table 1.

Table 1 Baseline demographic and clinical characteristics of enrolled patients.

Variable	Levels	Training group (N = 130)	Test group (N = 54)	Total (N = 184)	p	
Outcome	No portal vein thrombosis	90 (69.2%)	38 (70.4%)	128 (69.6%)	1	
	Portal vein thrombosis	40 (30.8%)	16 (29.6%)	56 (30.4%)		
Age	≤45	33 (25.4%)	11 (20.4%)	44 (23.9%)	0.592	
	>45	97 (74.6%)	43 (79.6%)	140 (76.1%)		
Sex	Female	61 (46.9%)	31 (57.4%)	92 (50%)	0.257	
	Male	69 (53.1%)	23 (42.6%)	92 (50%)		
BMI	<18.5	3 (2.3%)	2 (3.7%)	5 (2.7%)	0.29	
	18.5–24.9	88 (67.7%)	30 (55.6%)	118 (64.1%)		
	>25	39 (30%)	22 (40.7%)	61 (33.2%)		
Child-Pugh	A	120 (92.3%)	51 (94.4%)	171 (92.9%)	0.842	
	B	10 (7.7%)	3 (5.6%)	13 (7.1%)		
HV	Hepatitis B	82 (63.1%)	36 (66.7%)	118 (64.1%)	0.602	
	Hepatitis C	34 (26.2%)	15 (27.8%)	49 (26.6%)		
	Hepatitis A	1 (0.8%)	1 (1.9%)	2 (1.1%)		
	Alcoholic hepatitis	10 (7.7%)	1 (1.9%)	11 (6%)		
	Autoimmune hepatitis	3 (2.3%)	1 (1.9%)	4 (2.2%)		
SP	Laparoscopy	55 (42.3%)	30 (55.6%)	85 (46.2%)	0.152	
	Laparoscopy + pericardial vascular dissection	16 (12.3%)	7 (13%)	23 (12.5%)		
	Open	43 (33.1%)	9 (16.7%)	52 (28.3%)		
	Open + pericardial vessel disconnection	16 (12.3%)	8 (14.8%)	24 (13%)		
WBC	≤2.8	65 (50%)	25 (46.3%)	90 (48.9%)	0.767	
	>2.8	65 (50%)	29 (53.7%)	94 (51.1%)		
PSWBC	≤12.91	83 (63.8%)	29 (53.7%)	112 (60.9%)	0.264	
	>12.91	47 (36.2%)	25 (46.3%)	72 (39.1%)		
NLR	≤2.2	34 (26.2%)	17 (31.5%)	51 (27.7%)	0.579	
	>2.2	96 (73.8%)	37 (68.5%)	133 (72.3%)		
dNLR	≤1.8	56 (43.1%)	31 (57.4%)	87 (47.3%)	0.107	
	>1.8	74 (56.9%)	23 (42.6%)	97 (52.7%)		
LMR	≤1.84	26 (20%)	19 (35.2%)	45 (24.5%)	0.046	
	>1.84	104 (80%)	35 (64.8%)	139 (75.5%)		
PLR	≤62.83	51 (39.2%)	19 (35.2%)	70 (38%)	0.728	
	>62.83	79 (60.8%)	35 (64.8%)	114 (62%)		
SII	≤145.2	75 (57.7%)	29 (53.7%)	104 (56.5%)	0.739	
	>145.2	55 (42.3%)	25 (46.3%)	80 (43.5%)		
PSNLR	≤14.75	34 (26.2%)	18 (33.3%)	52 (28.3%)	0.421	
	>14.75	96 (73.8%)	36 (66.7%)	132 (71.7%)		
PS	≤4.05	70 (53.8%)	32 (59.3%)	102 (55.4%)	0.61	
	>4.05	60 (46.2%)	22 (40.7%)	82 (44.6%)		
PSLMR	≤0.03	56 (43.1%)	30 (55.6%)	86 (46.7%)	0.167	
	>0.03	74 (56.9%)	24 (44.4%)	98 (53.3%)		
PSPLR	≤64.37	52 (40%)	18 (33.3%)	70 (38%)	0.496	
	>64.37	78 (60%)	36 (66.7%)	114 (62%)		
PSSI	≤497.5	44 (33.8%)	12 (22.2%)	56 (30.4%)	0.166	
	>497.5	86 (66.2%)	42 (77.8%)	128 (69.6%)		
CAR	≤0.03	52 (40%)	30 (55.6%)	82 (44.6%)	0.077	
	>0.03	78 (60%)	24 (44.4%)	102 (55.4%)		
FAR	≤2.47	80 (61.5%)	32 (59.3%)	112 (60.9%)	0.902	
	>2.47	50 (38.5%)	22 (40.7%)	72 (39.1%)		
PVB	≤78.87	54 (41.5%)	23 (42.6%)	77 (41.8%)	1	
	>78.87	76 (58.5%)	31 (57.4%)	107 (58.2%)		
PCT	≤0.04	48 (36.9%)	20 (37%)	68 (37%)	1	
	>0.04	82 (63.1%)	34 (63%)	116 (63%)		
IL6	≤13.61	78 (60%)	38 (70.4%)	116 (63%)	0.246	
	>13.61	52 (40%)	16 (29.6%)	68 (37%)		
APTT	≤32.42	38 (29.2%)	13 (24.1%)	51 (27.7%)	0.596	
	>32.42	92 (70.8%)	41 (75.9%)	133 (72.3%)		
PT	≤13.95	58 (44.6%)	27 (50%)	85 (46.2%)	0.614	
	>13.95	72 (55.4%)	27 (50%)	99 (53.8%)		
INR	≤1.27	91 (70%)	39 (72.2%)	130 (70.7%)	0.902	
	>1.27	39 (30%)	15 (27.8%)	54 (29.3%)		
	>575	41 (31.5%)	20 (37%)	61 (33.2%)		
SPT	≤58.55	62 (47.7%)	31 (57.4%)	93 (50.5%)	0.299	
	>58.55	68 (52.3%)	23 (42.6%)	91 (49.5%)		
DPV	≤14.93	54 (41.5%)	24 (44.4%)	78 (42.4%)	0.842	
	>14.93	76 (58.5%)	30 (55.6%)	106 (57.6%)		
DSV	≤11.77	58 (44.6%)	25 (46.3%)	83 (45.1%)	0.963	
	>11.77	72 (55.4%)	29 (53.7%)	101 (54.9%)		
D-II	≤0.64	75 (57.7%)	31 (57.4%)	106 (57.6%)	1	
 	>0.64	55 (42.3%)	23 (42.6%)	78 (42.4%)	 	

Logistic regression analysis

Both univariate and multivariate logistic regression analyses were conducted on the entire cohort. In the univariate analysis (Table 2), a total of 16 variables with a significance level of P < 0.1 were selected for further investigation in the multivariate analysis. Subsequently, as it shown in Table 3, six variables emerged as independent risk factors for the occurrence of PVT. These factors include postoperative NLR (odds ratio (OR): 4.06, 95% confidence interval (CI) [1.43–12.77], P = 0.012), postoperative dNLR (OR: 2.66, 95% CI [1.11–6.71], P = 0.031), C-reactive protein to albumin ratio (CAR) (OR: 2.68, 95% CI [1.17–6.48], P = 0.023), portal vein diameter (DPV) (OR: 6.80, 95% CI [2.78–18.5], P < 0.001), platelet change value (PVB) (OR: 2.71, 95% CI [1.17–6.58], P = 0.023), and D-dimer (OR: 9.82, 95% CI [4.13–25.96], P < 0.001). A forest plot (Fig. 3) visually depicted the six independent risk factors.

Table 2 Univariate logistic regression analyses.

Variable	Levels	No portal vein thrombosis (N = 128)	Portal vein thrombosis (N = 56)	Total (N = 184)	p	
WBC	≤2.8	54 (42.2%)	36 (64.3%)	90 (48.9%)	0.009	
	>2.8	74 (57.8%)	20 (35.7%)	94 (51.1%)		
PSWBC	≤12.91	83 (64.8%)	29 (51.8%)	112 (60.9%)	0.132	
	>12.91	45 (35.2%)	27 (48.2%)	72 (39.1%)		
NLR	≤2.2	39 (30.5%)	12 (21.4%)	51 (27.7%)	0.279	
	>2.2	89 (69.5%)	44 (78.6%)	133 (72.3%)		
dNLR	≤1.8	64 (50%)	23 (41.1%)	87 (47.3%)	0.339	
	>1.8	64 (50%)	33 (58.9%)	97 (52.7%)		
LMR	≤1.84	35 (27.3%)	10 (17.9%)	45 (24.5%)	0.234	
	>1.84	93 (72.7%)	46 (82.1%)	139 (75.5%)		
PLR	≤62.83	54 (42.2%)	16 (28.6%)	70 (38%)	0.113	
	>62.83	74 (57.8%)	40 (71.4%)	114 (62%)		
SII	≤145.2	67 (52.3%)	37 (66.1%)	104 (56.5%)	0.117	
	>145.2	61 (47.7%)	19 (33.9%)	80 (43.5%)		
PSNLR	≤14.75	43 (33.6%)	9 (16.1%)	52 (28.3%)	0.024	
	>14.75	85 (66.4%)	47 (83.9%)	132 (71.7%)		
PS	≤4.05	77 (60.2%)	25 (44.6%)	102 (55.4%)	0.074	
	>4.05	51 (39.8%)	31 (55.4%)	82 (44.6%)		
PSLMR	≤0.03	66 (51.6%)	20 (35.7%)	86 (46.7%)	0.068	
	>0.03	62 (48.4%)	36 (64.3%)	98 (53.3%)		
PSPLR	≤64.37	54 (42.2%)	16 (28.6%)	70 (38%)	0.113	
	>64.37	74 (57.8%)	40 (71.4%)	114 (62%)		
PSSI	≤497.5	43 (33.6%)	13 (23.2%)	56 (30.4%)	0.217	
	>497.5	85 (66.4%)	43 (76.8%)	128 (69.6%)		
CAR	≤0.03	63 (49.2%)	19 (33.9%)	82 (44.6%)	0.079	
	>0.03	65 (50.8%)	37 (66.1%)	102 (55.4%)		
FAR	≤2.47	85 (66.4%)	27 (48.2%)	112 (60.9%)	0.031	
	>2.47	43 (33.6%)	29 (51.8%)	72 (39.1%)		
PVB	≤78.87	64 (50%)	13 (23.2%)	77 (41.8%)	0.001	
	>78.87	64 (50%)	43 (76.8%)	107 (58.2%)		
PCT	≤0.04	53 (41.4%)	15 (26.8%)	68 (37%)	0.085	
	>0.04	75 (58.6%)	41 (73.2%)	116 (63%)		
IL6	≤13.61	75 (58.6%)	41 (73.2%)	116 (63%)	0.085	
	>13.61	53 (41.4%)	15 (26.8%)	68 (37%)		
APTT	≤32.42	45 (35.2%)	6 (10.7%)	51 (27.7%)	0.001	
	>32.42	83 (64.8%)	50 (89.3%)	133 (72.3%)		
PT	≤13.95	69 (53.9%)	16 (28.6%)	85 (46.2%)	0.003	
	>13.95	59 (46.1%)	40 (71.4%)	99 (53.8%)		
INR	≤1.27	98 (76.6%)	32 (57.1%)	130 (70.7%)	0.013	
	>1.27	30 (23.4%)	24 (42.9%)	54 (29.3%)		
SPT	≤58.55	79 (61.7%)	14 (25%)	93 (50.5%)	<.001	
	>58.55	49 (38.3%)	42 (75%)	91 (49.5%)		
DPV	≤14.93	66 (51.6%)	12 (21.4%)	78 (42.4%)	<.001	
	>14.93	62 (48.4%)	44 (78.6%)	106 (57.6%)		
DSV	≤11.77	68 (53.1%)	15 (26.8%)	83 (45.1%)	0.002	
	>11.77	60 (46.9%)	41 (73.2%)	101 (54.9%)		
D-II	≤0.64	88 (68.8%)	18 (32.1%)	106 (57.6%)	<.001	
 	>0.64	40 (31.2%)	38 (67.9%)	78 (42.4%)		

Table 3 Multivariate logistic regression analyses.

Variables	Estimate	SE	z	P	OR	95% CI	
PSNLR>14.75	1.40	0.56	2.52	0.01	4.06	1.43–12.77	
PSdNLR>4.05	0.98	0.46	2.15	0.03	2.66	1.11–6.71	
CAR>0.03	0.99	0.43	2.27	0.02	2.68	1.17–6.48	
PVB>78.87	1.00	0.44	2.27	0.02	2.71	1.17–6.58	
D-II>0.64	2.28	0.47	4.91	0.00	9.82	4.13–25.96	
DPV>14.93	1.92	0.48	4.00	0.00	6.80	2.78–18.50	

Figure 3 Forest plot of six independent risk factors.

Nomogram predicting the probability of PVT

The ROC curve of PVT constructed based on the six potential factors were presented in Fig. 4. The respective areas under the curve (AUC) of postoperative neutrophil-to-lymphocyte ratio (PSNLR) and postoperative derived granulocyte lymphocyte ratio (PSdNLR) were 0.563 and 0.57, the AUC of CAR and PVB were 0.556 and 0.587, moreover, the AUC of DPV and D-dimer were 0.703 and 0.683, these results showed that these indexes had certain diagnostic efficacy. A predictive nomogram was developed using postoperative neutrophil to lymphocyte ratio (NLR), postoperative derived granulocyte to lymphocyte ratio (dNLR), C-reactive protein to albumin ratio (CAR), D-dimer, portal vein diameter (DPV), and platelet change value (PVB). The probability of PVT can be estimated by summing the scores associated with each factor and aligning the total score with the corresponding probability on the bottom scale (Fig. 5). This nomogram provides a visual and practical tool for clinicians to assess the likelihood of PVT in post-splenectomy patients with cirrhosis and hypersplenism.

Figure 4 ROC curves of six predictive indicators.

Figure 5 A nomogram predicts the probability of portal vein thrombosis (PVT).

Using this nomogram, each patient’s specific value should be located on each variable axis, and a line is drawn upward to identify the point for each variable value. The sum of these points can be found on the “total score axis”, with the vertical line drawn downward determining the risk of PVT.

Model evaluation and validation

The ROC curve of the training cohort is shown in Fig. 6A, the AUC of the predictive model in the training cohort was 0.817, indicating that the model has good discriminative ability. In Fig. 6B, we can see that the calibration curves of internal (training cohort) was very close to the 45° oblique line, showing that there was a great consistency between the predicted and actual results. In the validation cohort, the ROC curve of the model was presented in Fig. 7A, the AUC was 0.876, and the models also showed highly satisfactory calibration capabilities predictive performance in the validation cohort (Fig. 7B). Moreover, decision curve analysis (DCA) shown in Fig. 8 indicated that the patients using our model can get more net benefit than the patients with complete intervention or no intervention at all. It means the model has potential clinical usefulness as a nomogram.

Figure 6 ROC and calibration curves of training cohort.

(A) The ROC curves of the nomogram predicting PVT in training cohort. (B) Calibration curve of prediction model in training cohort.

Figure 7 ROC and calibration curve of validation cohort.

(A) The ROC curves of the nomogram predicting PVT in validation cohort. (B) Calibration curve of prediction model in validation cohort.

Figure 8 The clinical decision-making of the model in the overall cohort (DCA) curve.

Discussions

It is well known that PVT is a high-risk complication in patients with cirrhosis and hypersplenism after splenectomy. Once PVT occurs after surgery, it may cause serious progressive liver problems such as jaundice, ascites, and cirrhosis, ischemic intestinal necrosis, followed by gastrointestinal symptoms such as abdominal pain, diarrhea, and vomiting, and abnormal coagulation function may increase the patient’s risk of thrombosis, including serious complications such as intravenous thrombosis and pulmonary embolism, which will significantly reduce the patient’s quality of life. Therefore, in clinical work, there is an urgent need to optimize the early detection of people at high risk of PVT after splenectomy. Compared to other PVT prognostic map (Li et al., 2022; Kinjo et al., 2010), the experimental results of this study provide new insights into the risk of PVT after splenectomy in patients with post-hepatitis cirrhosis. Our data analysis showed that postoperative neutrophil-lymphocyte ratio (NLR), postoperative derived neutrophil-lymphocyte ratio (dNLR), and preoperative leukocyte-platelet ratio (CAR), portal vein diameter, platelet changes value, and D-dimer are associated with the risk of thrombosis, each serving as an independent risk factor. In clinical practice, doctors can easily obtain indicators such as NLR and dNLR from routine blood tests, assessing PVT risk faster and more accurately. This is of great significance for clinical decision-making, especially in emergencies, where rapid assessment of PVT risk can greatly improve patient treatment outcomes and quality of life.

In recent years, with the deepening of research on systemic inflammation indicators in disease progression, this series of indicators is of great significance in guiding clinical work. Growing evidence demonstrated a strong correlation between systemic inflammatory markers and various diseases, such as thrombosis of venous thromboembolism (VTE), dementia, mood disorders, atrial fibrillation/flutter (AF), ventricular arrhythmias (VA), chronic atrial fibrillation, hepatocellular carcinoma, colorectal cancer and so on (Yang et al., 2023; Mouchli et al., 2021; Chen, Xin & Xu, 2022; Kang et al., 2021). In regard to thrombosis of VTE, terms such as “immune thrombosis” and “thromboinflammation” have been used to describe the reactions/mechanisms involved in thrombosis of VTE, innate immunity, and inflammatory factor storms (Guo & Rondina, 2019; Sharifian-Dorche et al., 2021; Mandel et al., 2022). However, most studies mainly focused on the correlation between systemic inflammatory markers and deep vein thrombosis, and there are relatively few researches on the correlation between systemic inflammatory markers and portal vein thrombosis after splenectomy for posthepatic cirrhosis. Therefore, the objective of this study was to explore the correlation between portal vein thrombosis and systemic inflammatory markers after splenectomy for posthepatic cirrhosis. In this study, we found that NLR, dNLR, CAR and other indicators are significantly associated with the risk of PVT. Systemic inflammation indicators include neutrophil-to-lymphocyte ratio (NLR, neutrophil/lymphocyte), lymphocyte-to-monocyte ratio (LMR, lymphocyte/monocyte), platelet-to-lymphocyte ratio (PLR, platelets/lymphocytes), systemic immune inflammatory index (SII, neutrophils × platelets/lymphocytes), derived neutrophil-lymphocyte ratio (dNLR), C-reactive protein to albumin ratio (CAR, C -reactive protein/albumin), fibrinogen to albumin ratio (FAR, fibrinogen/albumin), and so on. These indicators have been proven in previous studies to predict the relationship between inflammatory status and disease progression in different situations (Zhang et al., 2022; Mazza et al., 2018; Levi, van der Poll & Buller, 2004). The inflammatory response is not only the body’s natural defense mechanism against injury and infection, but may also lead to tissue damage and dysfunction. In the setting of cirrhosis and post-splenectomy, the inflammatory response may exacerbate the condition and increase the risk of complications, including PVT (Zhu et al., 2018; Ming et al., 2018). Our research shows that these indicators can serve as early warning signs of disease progression and severity.

In this study, first, high levels of postoperative NLR (OR: 4.06, 95% CI [1.37–12.06], P = 0.012) were significantly associated with PVT, emphasizing the key role of postoperative inflammatory response in this complication. In a study of venous thrombosis in polycythemia vera, Alessandra’s team found that NLR values ≥5 (HR = 2.13, p = 0.001) were independently associated with venous thrombosis (Carobbio et al., 2022), which is consistent with our research results. In general, neutrophils contribute to thrombus accumulation by releasing proteases, and coagulation factors, and serving as scaffolds for cell attachment and fibrin polymerization. Although the role of lymphocytes in acute thrombosis is unclear, lymphocytes are known to be involved in regulating innate immune cell recruitment and activity during thrombus resolution. A recently reported specialized Treg lymphocyte subset accumulates in venous thrombus and is critical for thrombus resolution (Zhang et al., 2020; Kondelkova et al., 2010). Based on these clinical findings, we can support the recent biological view that innate immune cells play a role in the process of venous thromboembolism, which is mainly manifested by an increase in neutrophils and a decrease in lymphocytes. Secondly, postoperative dNLR, CAR, and PVB were also identified as an independent risk factor, highlighting the complex interaction of systemic inflammatory status. C-reactive protein (CRP) is an acute inflammatory marker whose levels are often elevated in inflammatory or infectious states. Albumin is a protein with anti-inflammatory effects, the levels of which may be reduced in inflammatory states. Therefore, CAR is usually regarded as a comprehensive inflammatory status indicator, reflecting the degree of inflammatory activation in the body. In this study, due to the lack of clinical data, we only collected preoperative CAR. During the occurrence of PVT, the inflammatory state may play a key role in the early stage of thrombosis, and CAR can be used as a comprehensive manifestation of inflammatory indicators. Kaplangoray et al. (2023) found that in patients who underwent primary percutaneous coronary intervention (pPCI) due to ST-segment elevation myocardial infarction (STEMI), CRP caused the occurrence of coronary thrombosis, with the OR of 10.206. Patients with cirrhosis may be accompanied by a chronic inflammatory state due to liver disease (Kaplangoray et al., 2023). In this case, the increase in CAR may be more significant because the liver has a direct impact on protein metabolism and synthesis. Therefore, CAR may be more sensitive in patients with cirrhosis and can more accurately reflect their overall inflammatory status. Our study found CAR (OR: 2.68, 95% CI [1.14–6.28], P = 0.023), which means that for every unit increase in preoperative CAR, the risk of postoperative PVT formation will increase 2.68 times. The specific mechanism still needs further biological and clinical research to elucidate. In addition, portal vein diameter >14.93 mm, D-dimer >0.64 mg/L, and postoperative platelet change value >78.87 109/L were significantly positively correlated with PVT. Studies have shown that the larger the diameter of the portal vein, the higher the portal hypertension and the slower the portal vein flow velocity. However, a larger portal vein diameter is often associated with intravenous intimal damage. Therefore, an enlarged portal vein diameter will stimulate thrombosis regardless of the surgical approach (Lin et al., 2023). In addition, the DPV of this study is consistent with previous research results (Wu et al., 2022), the odds ratio (OR value) is 6.8, which means that for every unit increase in DPV, the change (increase) in PVT formation is 6.8 times. According to reports, the incidence rate of PVT after splenectomy in patients with cirrhosis is 9.8%–47.9% (Kinjo et al., 2010). Currently, the mechanism of PVT is still under study and may be related to hypercoagulable state, platelet activation, endothelial cell modification or hemodynamic changes (Lu et al., 2020). In most cases, the platelet, red blood cell, and white blood cell counts of patients with PH increase significantly within a short period after splenectomy, and the blood becomes hypercoagulable. Some studies have found that postoperative platelet change rate is closely related to PVT (odds ratio (OR): 1.78, 95% confidence interval (CI) [1.24−2.62], P = 0.002; OR: 1.43, 95% CI [1.16–1.77], P < 0.001) (Li et al., 2022), this study uses the platelet change value (postoperative platelet mean–preoperative platelet count) as a predictive index, the odds ratio (OR value) is 2.37, consistent with the above point of view, platelet changes reflect postoperative coagulation status, high platelet changes values may increase the risk of blood clots. It is well known that D-dimer is associated with thrombosis, and its elevation increases the risk of thrombosis (Huang, Yu & Peng, 2021; Qian & Li, 2017). Dai et al. (2015a) did a meta-analysis and found that D-dimer might be regarded as a diagnostic marker for PVT in liver cirrhosis. However, there are still a few studies that report that D-dimer might not be useful to identify the presence of portal venous system thrombosis (PVST) in liver cirrhosis (Dai et al., 2015b). The results of our study show that the D-dimer odds ratio (OR value) is 9.82, which is highly correlated with postoperative PVT formation, meaning that for every unit increase in D-dimer, the change (increase) in PVT formation is 9.82 times. The results suggest that D-dimer is a risk factor for PVT in cirrhotic patients following splenectomy, which is consistent with the study result of Dai et al. (2015a). Finally, although systemic inflammation markers have shown potential in predicting PVT risk, there are several limitations. First, inflammatory markers are affected by multiple factors, including infection, surgery, and other non-specific factors. Secondly, there is currently a lack of unified clinical standards for the interpretation of these indicators. Therefore, future studies are needed to validate the utility of these indicators in larger patient populations and explore the possibility of combining these indicators with other biomarkers and clinical data.

In summary, there is currently no standard prevention protocol for PVT after splenectomy in patients with cirrhosis and hypersplenism. In recent years, most scholars have advocated early postoperative preventive anticoagulation therapy, which is more helpful in reducing the incidence of PVT. However, there is no definite answer as to what kind of patients are better served by preventive anticoagulation. What is exciting is that in this study, the accuracy of predicting PVT by a model based on systemic inflammation indicators was as high as 81%. This can effectively distinguish high-risk individuals of PVT, thereby guiding clinicians to take timely targeted individualized preventive measures. The decision curve indicated that the patients who had used this model can get more net benefit than the patients with complete intervention or no intervention at all. It means that the model has potential clinical usefulness as a nomogram. About the clinical utility of the study, via this predictive model, the clinicians can early screen and identify the high-risk patients of PVT with cirrhosis after splenectomy in the clinical work, which will be very useful to help doctors to stratify the high-risk patients of PVT and take early corresponding measures to reduce the incidence of PVT. In addition, comparing with existed clinical predictive models of PVT, the systemic inflammation indicators included in this clinical predictive model are easily obtained from routine blood draw results in clinical practice, greatly improving the applicability and convenience. Although our study provides valuable insights, it also has several limitations. First, our study was based on a retrospective data analysis from a single centre, so there may be sample selection bias. Secondly, uncommon preoperative factors that may affect the formation of PVT, such as liver function, portal vein flow velocity, splenic vein flow velocity, degree of liver cirrhosis, and other clinical data were not included in this study. In addition, due to the lack of clinical data, inflammation-related indicators such as postoperative 24h C-reactive protein (CRP), procalcitonin (PCT), and interleukin 6 (IL-6) could not be included in this study. Therefore, further prospective multicentre studies may help to verify our experimental results. Furthermore, our model still needs to be validated in a larger independent sample.

Supplemental Information

Supplemental Information 1 Raw data

Supplemental Information 2 Abbreviations

Abbreviations

PVT portal vein thrombosis

Child-Pugh liver function grade

SP surgical approach

WBC white blood cells

NLR neutrophil to lymphocyte ratio

PLR platelet to lymphocyte ratio

dNLR derived granulocyte to lymphocyte ratio

LMR lymphocyte to monocyte ratio

CAR c-reactive protein to albumin ratio

FAR fibrinogen to albumin ratio

PVB platelet change value

PCT procalcitonin

IL6 interleukin 6

PT prothrombin time

INR international normalized ratio

APTT activated partial thromboplastin time

SPT spleen thickness

DPV portal vein diameter

DSV splenic vein diameter

D-II D-dimer

SII the systemic immune inflammation index

PS postoperative

SE standard deviation

OR odds ratio

CI confidence interval

Additional Information and Declarations

Competing Interests

Author Contributions

Human Ethics

Ethics

Data Availability

The authors declare there are no competing interests.

Xin Deng conceived and designed the experiments, performed the experiments, analyzed the data, prepared figures and/or tables, authored or reviewed drafts of the article, and approved the final draft.

Wenyan Liao conceived and designed the experiments, performed the experiments, analyzed the data, prepared figures and/or tables, authored or reviewed drafts of the article, and approved the final draft.

Xinmiao Jiang performed the experiments, analyzed the data, prepared figures and/or tables, authored or reviewed drafts of the article, and approved the final draft.

Shun Tu performed the experiments, analyzed the data, prepared figures and/or tables, authored or reviewed drafts of the article, and approved the final draft.

Xiangmin Xie performed the experiments, analyzed the data, authored or reviewed drafts of the article, and approved the final draft.

Yuji Xiao performed the experiments, analyzed the data, authored or reviewed drafts of the article, and approved the final draft.

Wuyao Chen performed the experiments, analyzed the data, authored or reviewed drafts of the article, and approved the final draft.

Huan Zeng performed the experiments, analyzed the data, authored or reviewed drafts of the article, and approved the final draft.

Chengming Ding conceived and designed the experiments, prepared figures and/or tables, authored or reviewed drafts of the article, and approved the final draft.

The following information was supplied relating to ethical approvals (i.e., approving body and any reference numbers):

Ethical approval for this study was obtained from the Medical Ethics Committee of The First Affiliated Hospital of the University of South China (NO. 2023ll1017001).

The following information was supplied relating to ethical approvals (i.e., approving body and any reference numbers):

The Medical Ethics Committee of The First Affiliated Hospital of the University of South China.

The following information was supplied regarding data availability:

The raw data is available in the Supplemental File.

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
