# Peer review of "Establishment and validation of systemic inflammatory index model and risk assessment of PVT in cirrhosis after splenectomy—a retrospective study"

_PeerJ, doi:10.7717/peerj.19254_

## Round 0.1 · original submission · Major Revisions

· Academic Editor

Major Revisions

It is recommended that the manuscript be revised in accordance with the observations and suggestions made by the reviewers, with particular attention to the comprehensive additional comments and fundamental reports presented by Reviewer 2.

Best regards,

·

Basic reporting

Study focuses on an interesting filed about plasma and instrumental markers able in providing risk for acute venous thrombosis after splenectomy in patients having high risk for thrombosis. Authors provided results to focus on the hypothesy of their research.
Authors well investigated and the methods were detailed described.

Experimental design

Authors provided a clear and robust statistical methodology.
Authors considered a number of markers as putative in suspecting the risk for development of the venous thrombosis of the portal vein.

Validity of the findings

I mean that this last is the most relevant finding from the research.
In my opinion, results from the study could to focus on not complete assessed question concerning the predictive test able in determining the risk for acute thrombotic events in patients having most severe disease.

Additional comments

However, retrospective analysis could not to achieve data from the real life.
On the other hand, data from consecutive enrolled patients will provided most effective findings to focus on this relevant clinica question.
In my opinion, this last rapresents the crude limit of this interesting and well conducted study

I mean tha the conclusions from the study are well stated.
Finally, I will be honored if Authors will consider results from published study doi: 10.3892/ijmm.2023.5255.

Reviewer 2 ·

Basic reporting

See my comments below.

Experimental design

See my comments below.

Validity of the findings

See my comments below.

Additional comments

The paper should be extensively revised.
1. In the “Objective” part of “ABSTRACT” section, there is an extra full stop at the end of sentence “…… utilizing these markers to detect high-risk individuals early on”.
2. The authors should keep the font uniform, such as “aforementioned” and “demonstrated” in the “Result” section and “cirrhotic” in the “Conclusion” section.
3. In the text, “D-II polymer” should be revised as “D-dimer”. The authors should use proper nouns. Additionally, there are some similar articles regarding association between D-dimer level and portal venous system thrombosis in liver cirrhosis (PMID: 26629017 PMID: 26021776), but they are missed. The authors can make some discussion about this controversial issue.
4. In the “Conclusion” part of the “ABSTRACT” section, the authors said “The predictive evaluation of our clinical prediction model is accurate and effective”. It is meaningless and obscure.
5. In the “Keywords” section, the authors should change “cirrhosis ,” to “cirrhosis,”.
6. The authors should standardize the use of full names and abbreviations. In the “Introduction” section, the authors said “Portal vein thrombosis (PVT) is generally defined as thrombosis in the main portal vein or its branches”. The authors should change “Portal vein thrombosis (PVT)” to “PVT”.
There are similar problems in the manuscript. The authors need to revise them.
7. In the “Introduction” section, the authors PMID: 27432511
8. In the “Data Collection” section, the authors should keep the font uniform, such as “attributes” and “electronic health records system of our hospital”.
9. In the “Data Collection” section, there is an extra full stop at the end of sentence “The clinical attributes of all patients were retrieved from the electronic health records system of our hospital”.
10. In the “Data Collection” section, the author should keep the writing style of exponents consistent. For example, the authors said “……including white blood cells (WBC) with a reference interval of 3.5~9.5×10^9/L” and “……neutrophil count (N) with a reference interval of 1.8~6.30 × 109/L”.
11. In the “Data Collection” section, the authors said “The preoperative Child-Pugh classification, excluding Class C, was divided into three levels: A, B, and C”, but the sentence was unclear.
12. In the “Variable Definitions” section, the authors said “Furthermore, routine ultrasound and CT scans were performed between 3 to 5 days, ……”, in which the word “between” should be revised as “from”.
13. In the “Model Verification” section, the authors said “Calibration curves, assessed using the "ResourceSelection" R package……”, in which the word “ResourceSelection” should be revised as “Resource Selection” or “Resource-Selection”.
14. In the “Logistic Regression Analysis” section, the authors should check the data carefully. The authors said “These factors include postoperative NLR [odds ratio (OR): 4.06, 95% confidence interval (CI): 1.37-12.06, P = 0.012], ……”, where the data about CI is inconsistent with the data in the Figure 3.
15. In the “Logistic Regression Analysis” section, the authors said “A forest plot (Figure 4) visually depicts the six independent risk factors”, where “Figure 4” should be revised as “Figure 3”.
16. In the “Model evaluation and validation” section, the authors said “In the validation cohort, the ROC curve of the model is presented in Figure 7A, the AUC was 0.817, ……”, where “0.817” should be revised as “0.876” in order to make the data in the text consistent with the data in the Figure 7A.
17. In the “Discussions” section, the authors said “……the key role of postoperative inflammatory response in this complication. effect ”. What is the intention of the “effect”?
18. In the “Discussions” section, the authors said “……CRP was closely related to coronary thrombosis, and CRP (OR: 10.206; 95% CI: 2.987-34.872, P < .001))”, but the sentence is also unclear.
19. In the “Discussions” section, the authors said “……the risk of postoperative PVT formation will increase 2.68 times,”. “,” after “time” should be revised as “.”.
20. In the “Discussions” section, the authors said “When the model predicts that the patient's risk of portal vein thrombosis is higher than 33% if intervention measures are taken immediately” and “Clinical benefit, if it is lower than 33%, no intervention measures can be taken temporarily, ……”, but the two sentences are unclear to express.
21. In the exclusion criteria of the Figure 1, the authors repeatedly described this exclusion criterion “Previous splenic embolization or anticoagulant (n=23)”.
22. The authors should further discuss the clinical significance of this study in this field.

Reviewer 3 ·

Basic reporting

1. The first paragraph of the introduction on the background of cirrhosis could be more concise as you have focused on PVT. A more concise presentation of the research gaps and objectives will improve the introduction.
2. Consider replacing the list of inflammatory factors with adding a brief explanation of why inflammatory factors have something to do with PVT, e.g. underlying mechanisms.
3. Line 68-71: It is not appropriate to mention that an indicator is a risk factor, such as DPV, but to state that increased DVP is a risk factor for PVT, other indicators you mentioned are the same.
4. In Tables 1 and 2, there is no explicit explanation of the criteria used to classify the levels and, in general, continuous variables are presented as average or medium numerical values under different conditions.
5. The discussion should be enriched by comparisons with previous studies to better contextualize the findings, rather than simply extending the results. It would be more interesting to add explanations of the differences and why you chose these inflammatory factors.

Experimental design

No comment.

Validity of the findings

It is recommended to include the OR and the 95% CI for the logistic regression in Table 2.

Additional comments

1. Note the spelling of “splenectomy” in the title.
2. Note the expression of “post-hepatitis cirrhosis” in line 96-97.
3. It would be better to list the full names of abbreviations, and only give the full name the first time the abbreviation appears in the article, rather than giving both abbreviations and full names whenever they appear.
4. The theme fonts in lines 122-123 are not consistent.

---

## Round 0.2 · Minor Revisions

· Academic Editor

Minor Revisions

Dear Dr Ding,
To proceed with your manuscript, you are requested to review the suggestions and corrections sent by the reviewers. Following this, you are required to resubmit your manuscript.
Yours sincerely

·

Basic reporting

Authors provide a clear english language .

References are consistent to prrovide helpful background.

Article and addtional figures are professional strctured

Experimental design

The research meets the journal's objectives.

Original primary research within Aims and Scope of the journal

Article debates on no defiined question that in turn is most relevant to better knowledge and
to reduce consequent gap.

Methods described with sufficient detail and information to replicate.

Validity of the findings

Results fron stidy seems to impact on recent questions on the field
.
Study was conducted by using a correct and complete general and statistical evaluation providing sound conclusive.

Study shows a clear link between the relased data with conclusive remarks

Additional comments

Authors are requested to provide more data on current, mechanism able in provoking the PVT.
Authors are requested to give more information concerning role played by inflammatory biomarkers on
evaluating both the risk monitoring and/or re-occurrence of the venous thromboembolism.

Reviewer 2 ·

Basic reporting

Comments to the authors
1. In the “Introduction” section, the authors said “Therefore, PVT is a noteworthy complication associated with splenectomy in PVT”, where the words “in PVT” are redundant.
2. The authors should further elaborate on the follow-up time and results.
3. In the “Demographics and Characteristics of Enrolled Patients” part of the “Results” section, there is a redundancy in the presentation regarding the PVT incidence rates.
4. In the “Discussion” section, the authors said “Our data analysis showed that postoperative neutrophil-lymphocyte ratio (NLR), postoperative derived neutrophil-lymphocyte ratio (dNLR)……”, the authors should only use their abbreviations. And there are similar problems in this manuscript, the authors need to revise them.
5. The authors should refine the contents of the "Discussion" section. Some contents are duplicated.

Experimental design

See my comments above.

Validity of the findings

See my comments above.

Additional comments

See my comments above.

Reviewer 3 ·

Basic reporting

no comment

Experimental design

no comment

Validity of the findings

no comment

Additional comments

no comment

---

## Round 0.3 · accepted · Accept

· Academic Editor

Accept

Dear author,

The last corrections have been accepted, and your manuscript is accepted. Congratulations on your work, and thank you for submitting it to PeerJ.